Reference gene selection for qRT-PCR assays in Stellera chamaejasme subjected to abiotic stresses and hormone treatments based on transcriptome datasets

Liu Xin 1 2
Guan Huirui 1 2
Song Min 1 2
Fu Yanping 1 2
Han Xiaomin 1 2
Lei Meng 1 2
Ren Jingyu 1 2
Guo Bin 1 2
He Wei 1 2
Wei Yahui 1 2 weiyahui@nwu.edu.cn
1 College of Life Science, Northwest University , Xi’an, Shaanxi , China
2 Key Laboratory of Resource Biology and Biotechnology in Western China, Northwest University , Xi’an, Shaanxi , China
Ribeiro-Barros Ana
Electronic publication date: 2018 Apr 3
Publication date: 2018
Volume: 6
Electronic Location ID: e4535
Received 2017 Dec 8; Accepted 2018 Mar 6
Copyright: © 2018 Liu et al.
Copyright year: 2018
Copyright holder: Liu et al.
License: This is an open access article distributed under the terms of the Creative Commons Attribution License, which permits unrestricted use, distribution, reproduction and adaptation in any medium and for any purpose provided that it is properly attributed. For attribution, the original author(s), title, publication source (PeerJ) and either DOI or URL of the article must be cited.
License URL: https://creativecommons.org/licenses/by/4.0/

Keywords: Reference gene, qRT-PCR, Stellera chamaejasme, Abiotic stresses, Hormone treatments, Transcriptome

Funding: Special Fund for Agro-scientific Research in The Public Interest 201203062 Scientific research plan projects of the Education Department of Shaanxi Provincial Government 2010JS091 Scientific Research from Shaanxi Provincial Department of Education 16JK1756 Natural Science Foundation of China 31702159, 81303159 Support Plan for Young Talents of Science and Technology Association of Colleges and Universities in Shaanxi Province 20150105 Bureau of science and technology in Shaanxi Province, Xi’an City, Beilin District GX1702 This work were supported by the Special Fund for Agro-scientific Research in the Public Interest (No. 201203062), the Scientific research plan projects of the Education Department of Shaanxi Provincial Government (No. 2010JS091), the Special Fund for Scientific Research from the Shaanxi Provincial Department of Education (No. 16JK1756), the Natural Science Foundation of China (No. 31702159; No. 81303159), the Support Plan for Young Talents of Science and Technology Association of Colleges and Universities in Shaanxi Province (No. 20150105), and the Bureau of Science and Technology in Shaanxi Province, Xi’an City, Beilin District (No. GX1702). The funders had no role in study design, data collection and analysis, decision to publish, or preparation of the manuscript.

==============================
Background

Stellera chamaejasme Linn, an important poisonous plant of the China grassland, is toxic to humans and livestock. The rapid expansion of S. chamaejasme has greatly damaged the grassland ecology and, consequently, seriously endangered the development of animal husbandry. To draft efficient prevention and control measures, it has become more urgent to carry out research on its adaptive and expansion mechanisms in different unfavorable habitats at the genetic level. Quantitative real-time polymerase chain reaction (qRT-PCR) is a widely used technique for studying gene expression at the transcript level; however, qRT-PCR requires reference genes (RGs) as endogenous controls for data normalization and only through appropriate RG selection and qRT-PCR can we guarantee the reliability and robustness of expression studies and RNA-seq data analysis. Unfortunately, little research on the selection of RGs for gene expression data normalization in S. chamaejasme has been reported.

Method

In this study, 10 candidate RGs namely, 18S, 60S, CYP, GAPCP1, GAPDH2, EF1B, MDH, SAND, TUA1, and TUA6, were singled out from the transcriptome database of S. chamaejasme, and their expression stability under three abiotic stresses (drought, cold, and salt) and three hormone treatments (abscisic acid, ABA; gibberellin, GA; ethephon, ETH) were estimated with the programs geNorm, NormFinder, and BestKeeper.

Result

Our results showed that GAPCP1 and EF1B were the best combination for the three abiotic stresses, whereas TUA6 and SAND, TUA1 and CYP, GAPDH2 and 60S were the best choices for ABA, GA, and ETH treatment, respectively. Moreover, GAPCP1 and 60S were assessed to be the best combination for all samples, and 18S was the least stable RG for use as an internal control in all of the experimental subsets. The expression patterns of two target genes (P5CS2 and GI) further verified that the RGs that we selected were suitable for gene expression normalization.

Discussion

This work is the first attempt to comprehensively estimate the stability of RGs in S. chamaejasme. Our results provide suitable RGs for high-precision normalization in qRT-PCR analysis, thereby making it more convenient to analyze gene expression under these experimental conditions.

Introduction

Stellera chamaejasme Linn (Thymelaeaceae), a perennial herb and dominant plant of grassland desertification, is native to the northern and southwestern regions in China (Tseng, 1999). The whole plant is toxic, and its main toxic component is chamaejasmin, which can poison and kill cattle, sheep, and other livestock (Shi & Wei, 2016). The rapid spread of S. chamaejasme speeds up the process of grassland desertification and also poisons a large number of livestock in pasturing areas, causing great damage and loss to the local grassland ecology and livestock husbandry (Shi & Wei, 2016). Thus, it is of fundamental importance to elucidate the mechanisms of the rapid spread and stress adaptation of S. chamaejasme. However, limited genome sequence information is available, which greatly hinders the study of stress functional genes, ultimately resulting in a slow advancement of prevention and control measures. For the above reasons, our group established local transcriptome data for S. chamaejasme seedlings at five different time points (300 mM NaCl treatment for 0, 3, 12, 24, and 72 h; three biological replicates) using the Illumina HiSeq 4000 sequencing platform. After transcriptome sequencing and data analysis, fragments per kilobase of exons per million fragments mapped (FPKM) converted from RSEM (RNA-Seq by expectation maximization) were used to estimate unigene expression, which in some cases led to a few false-positive results.

Quantitative real-time polymerase chain reaction (qRT-PCR) is one of the most widely applied technologies to detect the expression levels of selected genes in many different samples (Huggett et al., 2005) because of its relatively accurate quantification, simplicity, specificity, high sensitivity, and high throughput capacity (Qi et al., 2016; Wang et al., 2016a). In the relative quantitative method of qRT-PCR data processing, the choice of internal genes is particularly important, and small changes in reference gene (RG) stability will significantly influence the accuracy of the relative expression of target genes (Dheda et al., 2005). Generally speaking, an ideal RG should be an endogenous gene that does change in any of the tested tissues or under any of the experimental conditions (Derveaux, Vandesompele & Hellemans, 2010; Li et al., 2016a; Wang et al., 2016b). In cells, some endogenous housekeeper genes with consistent relative expression are often used as RGs (Taylor et al., 2016).

Housekeeping genes (HKGs) generally refer to a class of highly conserved genes that have basic functionality in biochemistry metabolism in organisms (Fiume & Fletcher, 2012) and are normally expressed at relatively constant rates across different tissues (Warrington et al., 2000; Paolacci et al., 2009). However, several studies have found that the expression levels of HKGs vary to different degrees based on tissues, developmental stages, or experimental conditions (Thellin et al., 1999; Nicot et al., 2005; Wu et al., 2016). Therefore, it is necessary to select stably expressed HKGs as RGs before they are used to normalize target gene expression by qRT-PCR (Guenin et al., 2009; Gong et al., 2016). Up to date, many HKGs, such as 18S ribosomal RNA (18S rRNA), 28S ribosomal RNA (28S rRNA), β-actin (ACT), elongation factor 1-alpha (EF1A), glyceraldehyde-3-phosphate dehydrogenase (GAPDH), α tubulin (TUA), β tubulin (TUB), polyubiquitin (UBQ), cyclophilin (CYP), SAND protein family (SAND), malate dehydrogenase (MDH), glyceraledehyde-3-phosphate dehydrogenase of plastid 1 (GAPCP1), and so on, have been used to conducted studies for evaluating their stability under different experimental conditions (Demidenko, Logacheva & Penin, 2011; Chen et al., 2015; Cao, Wang & Lan, 2016; Ferraz Dos Santos et al., 2016; Wang et al., 2017).

As of now, there is no available internal control gene for qRT-PCR data normalization in S. chamaejasme, so we were unable to verify transcriptome sequencing results, analyze the expression patterns of salt or stress-related genes, or further clarify its spread mechanism. To solve this problem, in our study, we selected 10 candidate RGs based on the local salt S. chamaejasme transcriptome database and then determined their expression profiles in five different stages under various abiotic stresses (drought, salt, and cold) and with three hormone treatments (abscisic acid, ABA; gibberellin, GA; ethephon, ETH) by qRT-PCR and further evaluated their expression stabilities using three popular software packages: geNorm (Vandesompele et al., 2002), NormFinder (Andersen, Jensen & Orntoft, 2004), and BestKeeper (Pfaffl et al., 2004). The 10 candidate genes were 18S, 60S, CYP, EF1B, GAPCP1, GAPDH2, MDH, SAND, TUA1, and TUA6. Two target genes, Delta 1-pyrroline-5-carboxylate synthetase 2 (P5CS2), which encodes a crucial enzyme in the proline synthesis pathway under stress conditions by activating glutamate 5-kinase and glutamate-5-semialdehyde dehydrogenase (Strizhov et al., 1997), and GIGANTEA (GI), a circadian regulated gene whose protein product has not only been shown to regulate photoperiodic flowering and various developmental processes but has also been implicated in mediating cold stress and salinity stress responses (Cao, Ye & Jiang, 2005; Penfield & Hall, 2009; Park, Kim & Yun, 2013; Li et al., 2017), were used to verify the selected RGs.

Materials and Methods

Plant materials and stress treatments

Stellera chamaejasme seeds were collected from Qilian, Qinghai province. After peeling, the seeds were treated with 98% H2SO4 for 9–11 min and were then rinsed for 30 min with running water and planted in individual pots (14.5 × 14.5 × 6.5 cm) filled with nutrition soil, vermiculite and perlite (6:1:1). Germinated seeds were grown seven weeks and were then transferred to nurseries potted with double-layered filter paper for three days of adaptation cultivation. All of the nursery pots were placed in an artificial climate chamber at a temperature of 25 ± 2 °C during the day and 15 ± 2 °C at night, with a relative humidity of 50–55% and illumination intensity of 300 μmol m−2s−1 (14/10 h, day/night). Three pots of seven-week-old seedlings (three biological replicates) with a consistent growth status for each group were chosen and treated with abiotic stresses and hormone treatments.

For drought and salt treatments, 20% PEG-6000 (w/v, Sangon, Shanghai, China) (Zhuang et al., 2015) and 300 mM NaCl (Sangon, Shanghai, China) (Wang et al., 2015) were applied to irrigate the seedlings, respectively. For cold stress, the seedlings in the nursery pots were shifted to another artificial climate chamber at 4 °C. For hormone treatments, the leaves were sprayed with 0.1 mM ABA (Reddy et al., 2016; Wan et al., 2017), 0.1 mM GA (Li et al., 2016b), or 1.5 mM ETH (Wu et al., 2016). Seedlings were irrigated or sprayed every 12 h during the course of the experiment. Complete seedlings were carefully collected at 0, 3, 12, 24, and 48 h after treatments; immediately frozen in liquid nitrogen; and stored at −80 °C refrigerator until total RNA isolation.

Total RNA isolation and first strand cDNA synthesis

Five random individual plants, approximately 100 mg of seedlings in each sample, were used for total RNA isolation with a TRNzol reagent kit (TIANGEN, Beijing, China). The concentration and 260/280 and 260/230 ratios of the RNA samples were detected with a NanoDrop ND-1000 Spectrophotometer (NanoDrop Technologies, Wilmington, DE, USA), and the integrity of all of the RNA samples was verified by 1.0% (w/v) agarose gel electrophoresis (AGE). Subsequently, for reverse transcription PCR (RT-PCR) and qRT-PCR, a total of 3.0 μg of RNA was DNase I (Ambion, Waltham, MA, USA) treated and purified and then used to synthesize first strand cDNA by reverse transcription (Roche, Basel, Switzerland) in a 20 μl reaction system. Finally, cDNA diluted 50-fold with ddH2O, was used as the template for PCR amplification.

Candidate RG selection and primer design

Ten candidate RGs from the local S. chamaejasme transcriptome database were selected by using local NCBI-blast (version 2.4.0+). The sequences of these genes were used to design the qRT-PCR primers using Primer 5.0, Oligo 7.60, and Beacon Designer 8.20 software with the following criteria: melting temperature ™ of 50–65 °C, primer lengths of 17–25 bp, GC contents of 45–55%, and product lengths of 90–300 bp. The specificity of all of the selected primer pairs was observed via RT-PCR using the cDNA of control groups at 0 h as the template, and each gene fragmentation was underpinned by 2.0% (w/v) AGE and sequenced to ensure its reliability.

RT-PCR and qRT-PCR analysis

To confirm the specificity of each primer that we designed, we performed RT-PCR in a 25 μl system using the Bio-Rad C1000 PCR system (Bio-Rad, Hercules, CA, USA). The reaction system was as follows: 2.5 μl of Ex Taq buffer, 2 μl of dNTPs, 0.125 μl of TaKaRa Ex Taq (TaKaRa, Beijing, China), 60 ng of cDNA template, 0.2 μM reverse primer, 0.2 μM forward primer, and ddH2O to 25 μl. The RT-PCR reaction parameters were: 95 °C for 3 min, 40 cycles at 95 °C for 30 s, 58 °C for 30 s, 72 °C for 20 s, and 72 °C for 5 min. The amplification products were evaluated by 2.0% (w/v) AGE. To further confirm that the amplicon corresponded to the target sequence, PCR products contained in the agarose gel were extracted using a TIANgel Midi Purification Kit (TIANGEN, Beijing, China) and then sequenced using the dideoxy chain-termination method by Sangon Biotech (Shanghai, China) Co., Ltd.

qRT-PCR reactions were carried out with the Fast Start Universal SYBR GreenMaster (Roche, Basel, Switzerland) on a Bio-Rad CFX96 Real-Time PCR system (Bio-Rad, Hercules, CA, USA) in accordance with the manufacturer’s instructions. Reactions were conducted at 95 °C for 3 min as an initial denaturation, followed by 40 cycles at 95 °C for 10 s, 58 °C for 10 s, and 72 °C for 20 s. The melting curves, ranging from 58 °C to 95 °C, were determined to check the specificity of the amplicons. In the negative control group, qRT-PCR was performed using water instead of cDNA as the template. Three technical replicates were analyzed for each biological sample, and the final Ct values for each set of samples were the average of three biological replicates. A total of 45 cDNA samples from five time points in the control groups were used to determine the mean amplification efficiency (E) of each primer pair with the LinRegPCR program (Ruijter et al., 2009; Zhuang et al., 2015; Vavrinova, Behuliak & Zicha, 2016; Wu et al., 2016).

Data analysis of gene expression stability

Three different types of statistical tools: geNorm (version 3.5), NormFinder (version 0.953), and BestKeeper (version 1.0), were applied to rank the expression stability of the RGs across all of the experimental sets. For geNorm and NormFinder, the raw Ct values calculated by the CFX equipment (Tm) software were converted into the relative quantities using the formula 2−ΔCt (ΔCt = each corresponding Ct value − lowest Ct value) for gene expression profiling. For BestKeeper, the raw Ct values and amplification efficiencies estimated by the LinRegPCR program were used to calculate the coefficient of variation (CV) and standard deviation (SD). The RG with the lowest CV ± SD value was identified as the most stable gene, and the RG with SD value greater than 1.0 was judged to be unstable and should be avoided for gene expression normalization (Guenin et al., 2009). geNorm software was also used to determine the proper RG numbers with pair wise variation (Vn/Vn+1, n refers to the RGs number) between two sequential normalization factors.

Validation of RGs

To test the accuracy of the results, the geometric mean from the sort results of geNorm, NormFinder, and BestKeeper in each subset were used to calculate the comprehensive ranking of the candidate genes. The smaller the comprehensive ranking results, the better the gene expression stability. Then, the combination of the top two best RGs, best ranked RG and worst ranked RG were used to standardize the expression of two target genes, i.e., P5CS2 and GI, under different experimental conditions. Furthermore, the expression levels of P5CS2 and GI under salt stress calculated by the combination of the top two best RGs were also compared with the FPKM values in the S. chamaejasme transcriptome database.

Results

Selection of candidate RGs and target genes

After comparing the reported RGs in other species with the local transcriptome database of S. chamaejasme using the local Blast program, 10 RGs and two target genes were chosen to perform the gene normalization studies. The results showed that the E value of each blast gene indicated high homology. The untranslated region of these full-length unigene sequences were used to design the specific primers for RT-PCR and qRT-PCR. The unigene ID, NCBI accession number, gene symbol, gene name, homolog locus of 10 candidate RGs and two target genes, and E value compared with those of the homologous genes are listed in Table 1.

Table 1 Description of candidate reference genes and target genes.

Unigene gene ID	Accession number	Gene symbol	Gene name	Homolog locus	E value	
>c73334.graph_c0	MG516523	18S	18S ribosomal RNA	AH001810	1e-105	
>c68075.graph_c0	MG516524	60S	60S ribosomal RNA	KJ634810	0.0	
>c71629.graph_c0	MG516525	CYP	Cyclophilin	JN032296	2e-123	
>c70757.graph_c0	MG516526	EF1B	Elongation factor 1-beta	XM_013599463	9e-138	
>c67520.graph_c0	MG516527	GAPCP1	Glyceraldehyde-3-phosphate dehydrogenase of plastid 1	NM_106601	0.0	
>c74212.graph_c0	MG516528	GAPDH2	Glyceraledehyde-3-phosphate dehydrogenase 2	KM370884	0.0	
>c70711.graph_c1	MG516529	MDH	Malate dehydrogenase	HQ449567	0.0	
>c72957.graph_c1	MG516530	SAND	SAND family protein	NM_128399	0.0	
>c60567.graph_c0	MG516531	TUA1	Alpha-tubulin 1	AT1G64740	0.0	
>c65147.graph_c0	MG516532	TUA6	Alpha-tubulin 6	AT4G14960	0.0	
>c57696.graph_c0	MG516533	P5CS2	Delta 1-pyrroline-5-carboxylate synthetase 2	AT3G55610	0.0	
>c73625.graph_c0	MG516534	GI	GIGANTEA	KR813315	0.0	

Verification of the primer specificity and qRT-PCR amplification efficiency

The specificity of each primer was tested by 2.0% AGE, sequencing and melting curves analysis, which provided the expected amplicon length (Fig. S1) and single peak melting curves (Fig. S2). The primer sequences, amplicon size, product Tm, amplification efficiencies, and other relevant information are given in Table 2. The amplification product length of PCR varied from 94 to 267 bp. The Tm for all PCR products spanned from 76.0 °C for MDH to 83.5 °C for GAPCP1. The E values of these genes were between 1.824 (MDH) and 1.930 (GAPDH2), and the linear correlation coefficients (R2) varied from 0.994 (SAND) to 0.998 (CYP). In conclusion, we had every reason to believe that all of these specificity and efficiency estimates of the amplification were reliable for further analysis.

Table 2 Selected candidate RGs and target genes, primers, and amplicon characteristics.

Name	Forward primer sequences (5′–3′)
Reverse primer sequences (5′–3′)	Amplicon size (bp)	Product Tm (°C)	E	R2	
18S	CTATCCAGCGAAACCACAG
CCCACTTATCCTACACCTCTC	122	81.5–82.0	1.918	0.996	
60S	TTGTTCGATAGCATCCGTCT
ATAAAAGCAAACAACGGAAGCA	170	78.0–78.5	1.836	0.997	
CYP	ACATAGTTTGAGGCAACCTAGCAGT
TACACCTTCGCAGACAGTCGTT	161	80.0	1.854	0.997	
EF1B	GCAGTGAACTCTCCCCAG
CCAAACAGGGCATAAAAGAAC	191	78.0–79.0	1.842	0.998	
GAPCP1	CCATTAGATCCGTCGCCTGTT
TTGTTGGTGGCACTTCTGTAGC	192	83.0–83.5	1.834	0.998	
GAPDH2	GTGAAACTGGTCTCCTGGTATG
AACCCAGGCAACGCTTATA	115	81.0	1.930	0.998	
MDH	CCGCGACTTTGAATAAGCCCAT
AACTCAAAATCCTCGTCCCCAA	94	76.0–76.5	1.824	0.997	
SAND	CCTGCCAAGATACAATCCCA
TTTGTGCTGCCCTAAACGAG	267	80.0–80.5	1.872	0.997	
TUA1	GGCACTTTCGAGTTTTCGC
CCAGCTTGTCCGATGTGAA	97	79.0–79.5	1.840	0.998	
TUA6	GAAGGAATGGAGGAAGGGGAG
CAAACACAAGAAAGCGACAAATAAG	165	81.5–82.5	1.837	0.997	
P5CS2	TGACTTTATACGGTGGACCAA
TCCTCTGTGACAACGCAAT	178	82.5–84.5	1.839	0.997	
GI	ATGATTACAGAAACGGAATTAACTCA
TAACTCCATGAAGTACCGACAGA	112	79.5–81.0	1.858	0.994	

Expression profiles of candidate RGs

Boxplot analysis of the Ct values of different RGs in all of the experimental samples was performed using origin 2017 software (Fig. 1). The results demonstrated that the mean Ct values of the 10 candidate RGs presented a relatively wide field, from 19.26 to 30.76. 60S showed the least expression variation, while 18S exhibited the highest variation, with the Ct values ranging from 15.58 to 22.59. Since the Ct values are negatively related to the gene expression levels, the smaller the Ct value, the higher the gene expression level. As Fig. 1 shows, 18S was the highest-expressed RG for its lowest mean Ct value (15.58), and GAPCP1 had the lowest expression level on account of its maximum mean Ct value (32.58).

Figure 1 Distribution of Ct values for 10 candidate RGs across all S. chamaejasme samples.

Lines across the boxes denote the medians. The box represents the 25th and 75th percentile. The top and bottom whisker caps depict the maximum and minimum values, respectively. The white and black dots represent mean Ct values and potential outliers, respectively.

Analysis of gene expression stability

geNorm analysis. geNorm calculates the gene expression stability measure M value as the average pairwise variation V for the RG and other tested RGs (Vandesompele et al., 2002). The smaller the M value, the more stable the gene, and vice versa. In our study, the M values of the 10 candidate RGs of S. chamaejasme calculated by geNorm software were below 1.5 in all of the experimental settings (Fig. 2), suggesting that these genes should be considered relatively stable. As described in Figs. 2A–2C, GAPCP1 and EF1B under drought stress, GAPCP1 and 60S under cold stress, and EF1B and 60S under salt stress were the most stable RGs with the lowest M values of 0.07, 0.05, and 0.19, respectively. At the same time, in the ABA (Fig. 2D), GA (Fig. 2E), and ETH (Fig. 2F) treatment groups, SAND and TUA6, TUA1, and CYP, GAPDH2 and 60S were considered to be the most stable genes with the lowest M values of 0.21, 0.22, and 0.19, respectively. In addition, for all of the sample sets (Fig. 2G), GAPCP1 and CYP were suggested to be two most stable RGs. On the contrary, 18S was the least stable gene in all of the sets except for ETH treatment, in which TUA6 was the least stable gene.

Figure 2 Average expression stability value (M) and ranking of the 10 RGs across all treatments calculated using geNorm.

(A) Drought stress. (B) Cold stress. (C) Salt stress. (D) ABA treatment. (E) GA treatment. (F) ETH treatment. (G) All samples. The least stable genes are listed on the left, while the most stable genes are exhibited on the right.

NormFinder analysis. NormFinder provides a stability value for each gene by analyzing expression data obtained through qRT-PCR, which is a direct measurement for estimating expression variation when the gene is used for normalization (Dheda et al., 2005). The orders based on the stability values calculated by NormFinder (Table 3) were similar to those determined by geNorm. The stability ranking results under cold stress and in the GA treatment subsets were completely consistent with the results determined through geNorm; meanwhile, TUA6 and 18S were the two least stable genes for ETH treatment and the rest of the treatments. For the cold stress group, GAPCP1 and EF1B were the two most stable RGs (also ranked first by geNorm). For the salt stress group, GAPCP1 and TUA1 were the two most stable RGs, which was different from the geNorm results. For all samples, ABA-treated and ETH-treated subsets, NormFinder suggested that GAPCP1 and 60S, TUA1 and SAND, GAPDH2 and 60S were the most stable RGs, respectively, which were not exactly the same as the geNorm analysis results.

Table 3 Expression stability of 10 candidate reference genes calculated by NormFinder.

Rank	Drought	Cold	Salt	ABA	GA	ETH	ALL	
1	GAPCP1	GAPCP1	GAPCP1	TUA1	TUA1	GAPDH2	GAPCP1	
Stability	0.025	0.015	0.089	0.071	0.075	0.048	0.028	
2	EF1B	60S	TUA1	SAND	CYP	TUA1	60S	
Stability	0.052	0.018	0.089	0.072	0.075	0.051	0.031	
3	60S	EF1B	SAND	TUA6	GAPCP1	GAPCP1	CYP	
Stability	0.069	0.060	0.237	0.109	0.096	0.135	0.032	
4	SAND	GAPDH2	60S	CYP	SAND	SAND	SAND	
Stability	0.074	0.076	0.284	0.159	0.103	0.149	0.065	
5	CYP	CYP	EF1B	GAPCP1	60S	60S	TUA1	
Stability	0.245	0.238	0.319	0.188	0.323	0.150	0.130	
6	TUA1	SAND	CYP	60S	TUA6	MDH	EF1B	
Stability	0.316	0.385	0.371	0.201	0.358	0.207	0.163	
7	TUA6	TUA6	MDH	MDH	GAPDH2	EF1B	MDH	
Stability	0.326	0.481	0.447	0.255	0.414	0.251	0.185	
8	GAPDH2	TUA1	TUA6	EF1B	EF1B	CYP	TUA6	
Stability	0.405	0.523	0.726	0.401	0.754	0.359	0.294	
9	MDH	MDH	GAPDH2	GAPDH2	MDH	18S	GAPDH2	
Stability	0.615	0.586	1.286	0.516	0.836	0.486	0.357	
10	18S	18S	18S	18S	18S	TU 6	18S	
Stability	0.999	1.093	1.748	1.272	0.965	0.497	0.556	

BestKeeper analysis. BestKeeper evaluates the RG expression stability by calculating the CV and SD of the average Ct values. A lower CV value indicates more stable RG expression (Guenin et al., 2009). As shown in Table 4, under drought stress and for all of the sample subsets, TUA1 had the lowest CV ± SD values of 0.52 ± 0.16 and 0.53 ± 0.16 and was considered to be the most stable RG. Under the cold stress condition and salt stress and ABA treatment subsets, EF1B, which had the lowest CV ± SD values of 1.16 ± 0.31, 1.35 ± 0.36, and 1.04 ± 0.27, respectively, was identified as the best RG. In the GA treatment subset, TUA6 had the lowest CV ± SD value of 0.82 ± 0.22 and was the most stable RG. In the ETH treatment subset, BestKeeper suggested that GAPDH2 was the most stable RG with the lowest CV ± SD value of 0.68 ± 0.18. Additionally, only a few genes had a SD value greater than 1.0, indicating that most of the candidate RGs were relatively stable. Except for the ETH treatment subset, the most unstable RG among all of the experimental settings was 18S, which was the same as the results of geNorm and NormFinder.

Table 4 Expression stability of 10 candidate reference genes calculated by BestKeeper.

Rank	Drought	Cold	Salt	ABA	GA	ETH	ALL	
1	TUA1	EF1B	EF1B	EF1B	TUA6	GAPDH2	TUA1	
CV ± SD	0.52 ± 0.16	1.16 ± 0.31	1.35 ± 0.36	1.04 ± 0.27	0.82 ± 0.22	0.68 ± 0.18	0.53 ± 0.16	
2	SAND	GAPCP1	GAPCP1	TUA6	60S	60S	EF1B	
CV ± SD	0.89 ± 0.27	1.21 ± 0.37	1.88 ± 0.58	1.06 ± 0.29	1.02 ± 0.26	1 ± 0.26	0.82 ± 0.22	
3	GAPDH2	TUA1	60S	60S	TUA1	TUA1	GAPCP1	
CV ± SD	1.07 ± 0.28	1.39 ± 0.41	1.93 ± 0.5	1.27 ± 0.32	1.76 ± 0.56	1.16 ± 0.35	0.91 ± 0.28	
4	GAPCP1	60S	CYP	CYP	CYP	GAPCP1	60S	
CV ± SD	1.24 ± 0.38	1.4 ± 0.36	2.1 ± 0.61	1.37 ± 0.39	1.78 ± 0.52	1.17 ± 0.35	0.93 ± 0.24	
5	EF1B	GAPDH2	TUA1	SAND	SAND	MDH	TUA6	
CV ± SD	1.27 ± 0.34	1.48 ± 0.38	2.38 ± 0.75	1.45 ± 0.43	1.81 ± 0.56	1.2 ± 0.33	0.99 ± 0.27	
6	60S	CYP	SAND	GAPDH2	GAPDH2	SAND	SAND	
CV ± SD	1.71 ± 0.45	1.77 ± 0.52	2.39 ± 0.74	1.76 ± 0.45	1.9 ± 0.51	1.21 ± 0.36	1.1 ± 0.33	
7	TUA6	TUA6	TUA6	MDH	GAPCP1	EF1B	CYP	
CV ± SD	1.86 ± 0.52	1.87 ± 0.49	2.65 ± 0.74	1.92 ± 0.53	1.92 ± 0.61	1.43 ± 0.37	1.11 ± 0.32	
8	CYP	MDH	MDH	TUA1	EF1B	TUA6	GAPDH2	
CV ± SD	2.07 ± 0.61	2.06 ± 0.57	2.87 ± 0.81	1.95 ± 0.58	3.14 ± 0.85	1.61 ± 0.44	1.26 ± 0.33	
9	MDH	SAND	GAPDH2	GAPCP1	MDH	CYP	MDH	
CV ± SD	3.32 ± 0.94	2.31 ± 0.7	5.02 ± 1.27	2.07 ± 0.63	3.17 ± 0.88	1.62 ± 0.46	1.71 ± 0.48	
10	18S	18S	18S	18S	18S	18S	18S	
CV ± SD	6.33 ± 1.29	8.32 ± 1.63	12.7 ± 2.45	8.26 ± 1.46	6.34 ± 1.32	2.91 ± 0.52	3.56 ± 0.69	

Determination of the optimal number of RGs

At the suggestion of the geNorm Service tool, the critical value Vn/Vn+1 to determine the optimal RG number for qRT-PCR normalization is 0.15, below which the inclusion of an additional RG is not required (Vandesompele et al., 2002). As Fig. 3 shows, the V2/3 values of all of the experimental groups were less than 0.15, which indicated that a two RG combination would be sufficient to use for normalization.

Figure 3 Pairwise variation (Vn/Vn+1) values analysis in all the seven experimental subsets calculated using geNorm.

The cut-off value to determine the optimal number of RGs for qRT-PCR normalization is 0.15.

Comprehensive stability analysis of RGs

Table 5 and Fig. 4 summarize and rank the determination results obtained from the geNorm, NormFinder, and BestKeeper programs. Based on the analysis, GAPCP1 and EF1B were the most stable RGs under three abiotic stresses; thus, TUA6 and SAND, TUA1 and CYP, GAPDH2 and 60S were the best RG combinations under the ABA, GA, and ETH treatments, respectively. Still, 18S was the most unstable RG under all of the experimental conditions.

Table 5 Expression stability ranking of the 10 candidate reference genes.

Method	1	2	3	4	5	6	7	8	9	10	
A. Ranking order under drought stress (better–good–average)	
geNorm	GAPCP1/EF1B	60S	SAND	TUA1	CYP	TUA6	GAPDH2	MDH	18S		
NormFinder	GAPCP1	EF1B	60S	SAND	CYP	TUA1	TUA6	GAPDH2	MDH	18S	
BestKeeper	TUA1	SAND	GAPDH2	GAPCP1	EF1B	60S	TUA6	CYP	MDH	18S	
Comprehensive ranking	GAPCP1	EF1B	SAND	TUA1	60S	GAPDH2	CYP	TUA6	MDH	18S	
B. Ranking order under cold stress (better–good–average)	
geNorm	GAPCP1/60S	EF1B	GAPDH2	CYP	SAND	TUA6	TUA1	MDH	18S		
NormFinder	GAPCP1	60S	EF1B	GAPDH2	CYP	SAND	TUA6	TUA1	MDH	18S	
BestKeeper	EF1B	GAPCP1	TUA1	60S	GAPDH2	CYP	TUA6	MDH	SAND	18S	
Comprehensive ranking	GAPCP1	EF1B	60S	GAPDH2	CYP	TUA1	SAND	TUA6	MDH	18S	
C. Ranking order under salt stress (better–good–average)	
geNorm	EF1B/60S	CYP	GAPCP1	TUA1	SAND	MDH	TUA6	GAPDH2	18S		
NormFinder	GAPCP1	TUA1	SAND	60S	EF1B	CYP	MDH	TUA6	GAPDH2	18S	
BestKeeper	EF1B	GAPCP1	60S	CYP	TUA1	SAND	TUA6	MDH	GAPDH2	18S	
Comprehensive ranking	EF1B	GAPCP1	60S	TUA1	CYP	SAND	MDH	TUA6	GAPDH2	18S	
D. Ranking order under ABA treatment (better–good–average)	
geNorm	MON 1/TUA 6	CYP	60S	TUA1	MDH	GAPCP1	EF1B	GAPDH2	18S		
NormFinder	TUA1	SAND	TUA6	CYP	GAPCP1	60S	MDH	EF1B	GAPDH2	18S	
BestKeeper	EF1B	TUA6	60S	CYP	SAND	GAPDH2	MDH	TUA1	GAPCP1	18S	
Comprehensive ranking	TUA6	SAND	CYP	TUA1	60S	EF1B	MDH	GAPCP1	GAPDH2	18S	
E. Ranking order under GA treatment (better–good–average)	
geNorm	TUA1/CYP	GAPCP1	SAND	60S	TUA6	GAPDH2	EF1B	MDH	18S		
NormFinder	TUA1	CYP	GAPCP1	SAND	60S	TUA6	GAPDH2	EF1B	MDH	18S	
BestKeeper	TUA6	60S	TUA1	CYP	SAND	GAPDH2	GAPCP1	EF1B	MDH	18S	
Comprehensive ranking	TUA1	CYP	TUA6	GAPCP1	60S	SAND	GAPDH2	EF1B	MDH	18S	
F. Ranking order under ETH treatment (better–good–average)	
geNorm	GAPDH2/60S	TUA1	SAND	GAPCP1	MDH	EF1B	CYP	18S	TUA6		
NormFinder	GAPDH2	TUA1	GAPCP1	SAND	60S	MDH	EF1B	CYP	18S	TUA6	
BestKeeper	GAPDH2	60S	TUA1	GAPCP1	MDH	SAND	EF1B	TUA6	CYP	18S	
Comprehensive ranking	GAPDH2	60S	TUA1	GAPCP1	SAND	MDH	EF1B	CYP	TUA6	18S	
H. Ranking order under ALL stress (better–good–average)	
geNorm	GAPCP1/CYP	60S	TUA1	EF1B	SAND	MDH	TUA6	GAPDH2	18S		
NormFinder	GAPCP1	60S	CYP	SAND	TUA1	EF1B	MDH	TUA6	GAPDH2	18S	
BestKeeper	TUA1	EF1B	GAPCP1	60S	TUA6	SAND	CYP	GAPDH2	MDH	18S	
Comprehensive ranking	GAPCP1	60S	TUA1	CYP	EF1B	SAND	TUA6	MDH	GAPDH2	18S	

Figure 4 Comprehensive ranking of candidate genes calculated by the geometric mean of three types of rankings (geNorm, NormFinder, and BestKeeper) in each subset.

(A) Drought stress. (B) Cold stress. (C) Salt stress. (D) ABA treatment. (E) GA treatment. (F) ETH treatment. (G) All samples.

Reference gene validation

As shown in Figs. 5 and 6, when the best RG combinations were used for performing normalization, the expression levels of P5CS2 and GI were affected by different treatments. A sustained increase in expression level of P5CS2 was observed after drought stress, and a peak point was observed at 48 h (Fig. 5A). A tendency of first increase, after downward, and then upward in the transcript level of P5CS2 appeared after cold and GA treatments (Figs. 5B and 5E). Additionally, upregulated expression of P5CS2 was observed after salt and ABA treatments, and reached the maximum value at 12 and 24 h following a decrease (Figs. 5C and 5D). Whereas, P5CS2 expression was first downregulated at 3 h after ETH treatment and then began to continuous increase, reaching the maximum at 48 h (Fig. 5F). The maximal expression levels of GI under drought (Fig. 6A), cold (Fig. 6B), salt (Fig. 6C), ABA (Fig. 6D), and ETH (Fig. 6F) treatments also appeared prominent changes, which were 4.66-fold, 29.22-fold, 2.10-fold, 6.45-fold, and 2.45-fold higher than those of the control group, while there was no significant difference under GA treatment (Fig. 6E).

Figure 5 Relative expression levels of P5CS2 under different experimental conditions normalized by the most stable RG combination, the most stable gene and the most unstable gene.

(A) Drought stress. (B) Cold stress. (C) Salt stress. (D) ABA treatment. (E) GA treatment. (F) ETH treatment. Bars represent the standard error from three biological replicates.

Figure 6 Relative expression levels of GI under different experimental conditions normalized by the most stable RG combination, the most stable gene and the most unstable gene.

(A) Drought stress. (B) Cold stress. (C) Salt stress. (D) ABA treatment. (E) GA treatment. (F) ETH treatment. Bars represent the standard error from three biological replicates.

Compared with the best RG combinations for normalization of P5CS2 and GI, similar expression patterns were obtained when the most stable single genes, GAPCP1 (drought and cold), EF1B (salt), TUA6 (ABA), TUA1 (GA), and GAPDH2 (ETH), were used for normalization under the above treatments. However, different expression patterns were generated and the expression levels of P5CS2 and GI were overestimated when the least stable gene, 18S, was selected as the RG for normalization.

In particular, as shown in Fig. 5C, under salt treatment, when the RG combination (GAPCP1 and EF1B) was selected for normalization, gene expression of P5CS2 gradually increased from 0 h, reached the maximum at 24 h, and then began to slightly decline at 48 h. In the same way, the expression levels of GI increased at first, then decreased and maintained a lower level until 48 h (Fig. 6C). The expression trends of P5CS2 and GI over the first 24 h were generally consistent with those of RNA-seq (Fig. S3), which further validated the accuracy and reliability of our experimental results.

Discussion

qRT-PCR is currently viewed as a powerful technique that can be used to quantify target gene expression. The accuracy of qRT-PCR directly depends on the stability of the internal genes used. The use of inappropriate RGs for normalizing qRT-PCR data will lead to deviations in the results (Shivhare & Lata, 2016). In this study, three programs, geNorm, NormFinder, and BestKeeper, were used to select optimum RGs for six different experimental conditions. The 10 potential RGs exhibited differential stability in response to different stresses. Taking ABA treatment as an example, in the experimental subset, geNorm software ranked SAND as the best RG, and NormFinder regarded TUA1 as the most stable RG. However, BestKeeper identified EF1B as the best RG according to its lowest CV value. This means that the three types of software generated different results, and a solution was not found. Our study carried out a comprehensive analysis and provided ultimate stability ordering results by ranking the geometric means of the three software analysis results, which is a common strategy for evaluating the expression stability of RGs reported in previous scientific papers.

EF1B catalyzes the exchange of guanosine diphosphate bound to the G-protein, elongation factor 1-alpha (EF1A), for guanosine triphosphate, an important step in the elongation cycle of protein biosynthesis. It has been considered to be one of the most stable RGs during drought and salt stresses (Wan et al., 2017) and other stress conditions (Ma et al., 2013). In our study, EF1B was ranked as one of the two best RGs under drought and salt stresses according to the comprehensive ranking results, which was the same as in Stipa grandis (Wan et al., 2017). In addition, EF1B performed a better expression stability under three abiotic stresses than those in the three hormone treatments. The results showed that there are no universal RGs that are stably expressed in all biological materials and/or under all trial conditions.

The expression stability of two homologous RGs, TUA1 and TUA6, were estimated in our study. According to the results, the stability ranking of TUA1 was always better than that of TUA6 under all conditions except the ABA treatment, under which TUA6 exhibited better expression stability. Nevertheless, it is notable that the homologous RGs showed different rank orders in each subset, and in most cases, TUA1 showed better expression stability than TUA6. Cordoba et al. (2011) found that TUA1 was one of the most suitable RGs under NaCl stress and 2,4-dichlorophenoxyacetic acid treatment in Hedysarum coronarium. However, Gimeno et al. (2014) suggested that TUA6 should be discard for normalization under drought stress, salt stress, cold and heat shock treatment, and flooding treatment in switchgrass. Although these reports indicated the expression stability of TUA1 and TUA6 in different species, respectively, but at present we have not found any reports of the simultaneous use of TUA1 and TUA6 in stability analysis in other species. Therefore, this does not mean that the expression stability of TUA1 in other species must be better than that of TUA6.

18S is a frequently used HKG and is widely used for normalization in qRT-PCR analysis. Wang et al. (2017) reported that 18S rRNA was one of the most stably expressed gene under diverse heavy metals stresses in tea plants; Huang et al. (2017) also found that 18S rRNA was the most stable gene under UV irradiation and hormonal stimuli in Baphicacanthus cusia. However, our analysis results suggested that 18S was the most unstable RG in all of the experiment groups because of its excessively high expression level. In comparison with the best RG combination and the most stable RG, when 18S was selected as a RG to validate the expression of the two target genes P5CS2 and GI, their expression patterns were significantly overestimated, which was consistent with the findings in Oxytropis ochrocephala (Zhuang et al., 2015) and rice (Bevitori et al., 2014).

Two target genes, P5CS2 and GI, were used to verify the stability of the selected RGs for gene expression normalization. Strizhov et al. (1997) stated that expression of Arabidopsis thaliana P5CS (AtP5CS) is root and leaf specific and can be regulated by salinity, drought, and ABA. The same experimental results were reproduced in our experiments. We also found that P5CS could be efficiently expressed during the later period of cold stress, which may be a supplement to previous findings. The induction mechanism remains to be further studied. Cao, Ye & Jiang (2005) reported that GI is induced by cold stress, but not by salt, mannitol, and ABA. By contrast, Park, Kim & Yun (2013) and Kim et al. (2013) claimed that GI, as a negative regulator, participated in the regulation of salt stress in Arabidopsis by interacting with salt overly sensitive 2. Moreover, Riboni et al. (2016) revealed that ABA affects flowering through two independent regulatory mechanisms: activation of GI and constant (CO) functions upstream of the florigen genes and down-regulation of the suppressor of overexpression of CO1 (SOC1) signaling. Our findings indicated that the gene expression of GI not only changed under salt stress and cold stress but also underwent a significant change under drought, ABA, and ETH treatments. We have reason to believe that these mechanisms will be revealed with future in-depth experiments.

There is no doubt that it is necessary to select suitable RGs and/or RG combinations for gene normalization studies to obtain more accurate and reliable results. Combined with all of the validation results above, we observed that, in most cases, P5CS2 and GI showed similar response patterns when normalized by the most stable RGs combinations, but some differences still emerged. Unfortunately, we could not tell which choice was better for normalization. However, to eliminate the small variations caused by technical protocols in qRT-PCR, two or more RGs are often required to correct for non-specific experimental variation (Thellin et al., 1999; Bustin et al., 2009). In this study, two RG combinations, whose V2/3 values were less than 0.15 across all of the experimental subsets and thus GAPCP1 and EF1B for drought stress, cold stress and salt stress, TUA6 and SAND for ABA treatment, TUA1 and CYP for GA treatment, and GAPDH2 and 60S for ETH treatment, were suggested for the accurate normalization of target gene expression.

Conclusion

This study represents the first attempt to comprehensively analyze the stability of RGs for use as internal controls in qRT-PCR analysis of target gene expression in S. chamaejasme under three abiotic stresses and three hormone treatments by combining results from three different methods. The results indicated that the stability of an identical gene was not exactly the same under different treatments, and the stability ranking of the RGs calculated by three parameters was not identical under the same treatment. As a result, it makes sense to carry out a comprehensive analysis of the results of the three procedures. Moreover, it may be a better choice to select a combination of two or more RGs as an effective internal control to further improve the accuracy and reliability of gene expression normalization under different stresses. In conclusion, this study provides a guideline to select a valid RG combination that can ensure more accurate qRT-PCR-based gene expression quantification and basic data to facilitate future molecular studies on gene expression in S. chamaejasme and other Thymelaeaceae species (Che et al., 2016).

Supplemental Information

Supplemental Information 1 2.0% agarose gel electrophoresis of PCR products for each of the ten RGs and two target genes.

2.0% (w/v) agarose low-melt agarose, 1×TAE gel buffer were supplied with 70 V for 1 h. A 100 bp DNA ladder was used to determine approximate sizes of PCR products. The amplified product showed the expected size and no primer dimers.

Click here for additional data file.

Supplemental Information 2 Melting curves analysis for ten RGs and two target genes.

One single peak was obtained in each amplification reaction.

Click here for additional data file.

Supplemental Information 3 FPKM values of P5CS2 and GI based on the transcriptome datasets of S. chamaejasme by RNA-seq. (A) P5CS2. (B) GI.

Bars represent the standard error from three biological replicates. The expression tendencies of P5CS2 and GI from 0 h to 24 h calculated by FPKM value based on transcriptome database were accordance with the results normalized by the most stable RGs combination or only the most reference gene.

Click here for additional data file.

Supplemental Information 4 cDNA sequence data.

Click here for additional data file.

The authors are particularly grateful to Jiakun Dai, Na Fan, Shilan Feng, and other members of our research group for their helpful comments/suggestions to improve the experimental design.

Additional Information and Declarations

Competing Interests

Author Contributions

DNA Deposition

Data Availability

The authors declare that they have no competing interests.

Xin Liu conceived and designed the experiments, performed the experiments, analyzed the data, prepared figures and/or tables, authored or reviewed drafts of the paper, approved the final draft.

Huirui Guan performed the experiments, analyzed the data, prepared figures and/or tables, authored or reviewed drafts of the paper, approved the final draft.

Min Song performed the experiments, authored or reviewed drafts of the paper, approved the final draft.

Yanping Fu analyzed the data, contributed reagents/materials/analysis tools, authored or reviewed drafts of the paper, approved the final draft.

Xiaomin Han performed the experiments, authored or reviewed drafts of the paper, approved the final draft.

Meng Lei performed the experiments, authored or reviewed drafts of the paper, approved the final draft.

Jingyu Ren performed the experiments, authored or reviewed drafts of the paper, approved the final draft.

Bin Guo conceived and designed the experiments, authored or reviewed drafts of the paper, approved the final draft.

Wei He conceived and designed the experiments, contributed reagents/materials/analysis tools, authored or reviewed drafts of the paper, approved the final draft.

Yahui Wei conceived and designed the experiments, contributed reagents/materials/analysis tools, authored or reviewed drafts of the paper, approved the final draft.

The following information was supplied regarding the deposition of DNA sequences:

Sequences of 10 reference genes and two target genes described in this study are accessible via GenBank accession numbers MG516523–MG516534 and are also found in the Supplemental Information.

The following information was supplied regarding data availability:

The raw data are provided in Supplemental Dataset Files.

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
