# Peer review of "Reference gene selection for qRT-PCR assays in Stellera chamaejasme subjected to abiotic stresses and hormone treatments based on transcriptome datasets"

_PeerJ, doi:10.7717/peerj.4535_

## Round 0.1 · original submission · Minor Revisions

· Academic Editor

Minor Revisions

Dear Dr Xin,

Thank you for your submission to PeerJ. Please address the reviewer comments as noted below.

·

Basic reporting

The English language should be improved to ensure that the readers can clearly understand your text. Some examples where the language could be improved are highlighted in the annotated PDF file. The current phrasing makes comprehension difficult. Typing errors should also be corrected.

Experimental design

The section of materials and methods raises several doubts that should be clarified, namely:
1- The authors should include in the text a justification for choosing these particular 10 candidate genes (RGs).
2- No justification is given for the use of the concentration of 300mM for salt treatment nor the percentage of 20% of PEG for the drought treatment. Have the authors done some previous studies that show that these concentrations are adequate to induce the stresses?
3- In the case of plants obtained from seeds in which the variability will certainly be high, three biological replicates may not be sufficient to guarantee a good sampling, four replicates should be the minimum. Please explain why you did not use a greater number of plants in each treatment.
4- As the authors point out gene expression is dependent on the organ or tissue. So why was the study done on “complete seedlings” instead of leaves, roots and stems separately?
5- Did the authors used any negative controls for de qRT-PCRs? It should be mentioned in the text.
6- Did the authors check for DNA contamination in the RNA samples? It should be mentioned in the text.
7- It is not clear which cDNA was used to confirm primers specificity (lines 135- 140). Please explain.
8- It is not clear which cDNA samples were used to determine primers efficiencies (lines 144- 152). Please explain.

Validity of the findings

The sections of results and discussion raises several doubts that should be clarified, namely:

1- In lines 197-198 the authors say:” Instead, GAPDH2 showed the lowest expression level….”. This in not in accordance with Figure 1. Please check.

2- It is not clear how the comprehensive stability analysis of RGs was made. Please clarify which methodology / tool used.

3- Please check other comments/suggestions in the annotated PDF file.

Additional comments

It should be emphasized the importance of the previous selection of RGs because only in this way we can guarantee the reliability and robustness of the of expression studies and RNA-seq data analysis.
However, in its current form, the manuscript shows several flaws. Still, the authors are encouraged to respond and to resolve the issues raised.

Reviewer 2 ·

Basic reporting

The manuscript needs language editing.

L93-L95: when introducing GI, its role in salt stress resistance (see, e.g., www.ncbi.nlm.nih.gov/pubmed/19542296) should also be mentioned

Lines 302-309 describe in detail why 18S is not a suitable RG. I think this is obvious from Figures 4, 5 and 6 and does not have to be included in the discussion.

L312: tissue-specific - for which tissue?

Line 318: The reference for the SOS pathway is not correct. And I wonder whether the authors wanted to quote www.ncbi.nlm.nih.gov/pubmed/23322040 for the effects of ABA on GI expression instead of www.ncbi.nlm.nih.gov/pubmed/23656866.

Format of References:italicize species names, do not capitalize titles

Experimental design

The paper fits the Aims and Scope of PeerJ, and I see no problems with the experimental design and statistical analysis. However, I wonder why not more candidates for RGs were examined. Besides, why was mon1 selected as a housekeeping gene? It encodes a protein that, among other functions, regulates programmed cell death (https://www.ncbi.nlm.nih.gov/pubmed/27799422).

Validity of the findings

The choice of 10 potential RGs led to the identification of suitably stable RG combinations for the six experimental conditions under analysis. However, as shown in Figures 5 and 6, only one RG combination was suited for more than one condition (drought/cold). This raises the question whether the analysis of more potential RGs would have yielded a combination that worked for more conditions.

Additional comments

The text has to be corrected by a native speaker.

---

## Round 0.2 · Minor Revisions

· Academic Editor

Minor Revisions

Dear Dr Xin,

Thank you to consider Peer J to publish your work. Altough the MS has improved considerably, I still think that some minor details should be addressed to improve the quality of the paper. These are highlighted in the attached pdf file. Please send me the final version as you feel appropriate, and after that I will accept the paper.

Sincerely,
Ana I Ribeiro-Barros

---

## Round 0.3 · accepted · Accept

· Academic Editor

Accept

Dear Dr Xin and Dr. Yahui,

After Reading the second version of the revised MS, it is my pleasure to communicate that the MS is now accepted for publication.
Thank you to consider Peer J to publish your work.

Sincerely
Ana I Ribeiro-Barros